# Photon bubble turbulence in cold atom gases

R. Giampaoli [1], João D. Rodrigues [1,2 ✉], José-António Rodrigues[1,3] & J. T. Mendonça [1]

Turbulent radiation flow is commonplace in systems with strong, incoherent, light-matter interactions. In astrophysical contexts, photon bubble turbulence is considered a key mechanism behind enhanced radiation transport, and its importance has been widely asserted for a variety of high energy objects such as accretion disks and massive stars. Here, we show that analogous conditions to those of dense astrophysical objects can be obtained in large clouds of cold atoms, prepared in a laser-cooling experiment, driven close to a sharp electronic resonance. By accessing the spatially-resolved atom density, we are able to identify a photon bubble instability and the resulting regime of photon bubble turbulence. We also develop a theoretical model describing the coupled dynamics of both photon and atom gases, which accurately describes the statistical properties of the turbulent regime. This study thus opens the possibility of simulating radiation-dominated astrophysical systems in cold atom experiments.

[1] Instituto de Plasmas e Fusão Nuclear, Instituto Superior Técnico, Universidade de Lisboa, Lisbon, Portugal. [2] Physics Department, Blackett Laboratory, Imperial College London, London, UK. [3] Departamento de Física, Universidade do Algarve, Campus de Gambelas, Faro, Portugal. ✉email: j.marques-rodrigues@imperial.ac.uk

In dense astrophysical systems, radiation often propagates diffusively instead of ballistically. It has been argued that condition may exist where the coupled dynamics of light and matter lead to a particular form of turbulence—photon bubble turbulence (PBT)[1–3]. Evidences of photon bubble instabilities have indeed been observed in high-mass binary pulsars[4]. Moreover, magnetized radiation-dominated atmospheres are known to be stirred by photon bubbling, fostering radiation transport and energy redistribution inside the medium[5,6]. Similar processes also seem to play a prominent role in the formation of massive stars[2] and cooling of accretion disks[7]. Photon bubble instabilities may also be related with the emergence of super-Eddington luminosity[8–10], or the formation of proto-planet and voids in dust clouds[11].

PBT consists of radiation-filled pockets (bubbles) that grow and percolate through the matter fluid. Bubbles eventually leave the medium or dissipate energy decaying into smaller structures. Thus, regions of high density alternate with regions almost devoid of particles and dominated by radiation pressure, forming a distinctive pattern, which turbulently propagates through the medium. Laboratory-based realizations of photon bubble instabilities would be invaluable to the understanding of processes occurring in such exotic astrophysical objects[12,13]. However, despite the existence of proposals using high-power lasers[14], no experimental results have been reported so far.

Here, we investigate photon bubble instabilities in cold atom gases. These instabilities occur when radiation and thermal pressure become of comparable strength, which can be tailored in cold atom experiments. While strong light–matter interactions in astrophysical objects are the result of high densities, in dilute cold atom gases, analogous conditions originate from the sharp electronic resonances and the sub-millikelvin operating temperatures. In particular, by tuning the cooling and trapping laser beams close to an electronic resonance, we observe an abrupt transition from a stable to a turbulent regime, the latter being characterized by large spatio-temporal atom density fluctuations. The statistical properties of the turbulent regime are characterized by measuring both the auto-correlation function and the power spectrum of the atom density fluctuations, revealing the presence of quasi-coherent structures originating from the local nucleation of photon bubbles. This interpretation is further supported by a theoretical model describing the coupled dynamics of both photon and atom gases.

## Results

**Experimental procedure**. Our experiment consists of a magneto-optical trap (MOT)[15] where around $10^9$ Rb$^{85}$ atoms are cooled and trapped at ~200 μK[16,17]—check 'Methods' and the Supplementary Information for further details on the experiment. Atoms are constantly driven by six independent cooling (and trapping) laser beams crossing at the centre of the trap. The laser frequency can be precisely tuned in the vicinity of the electronic resonance, controlling the effective coupling between atoms and photons. Direct access to the atom density profile is accomplished via the pump-probe set-up depicted in Fig. 1. This technique, further described in the Supplementary Information, circumvents line-of-sight integration effects, which become relevant only at scales smaller than the thickness of the atom slab, typically around 230 μm.

The trap is continuously maintained by the cooling and trapping beams, with a transition from a stable to a turbulent regime being observed as we bring the frequency of these lasers close to that of the electronic transition. We define the laser detuning as $\delta = \omega_L - \omega_a$, with $\omega_L$ and $\omega_a$ the laser and electronic transition frequencies, respectively. We load the

trap at different values of $\delta$, ranging from $-4\Gamma$ to $-0.75\Gamma$, with $\Gamma$ the transition linewidth. The atom cloud is continuously driven for a given time interval $\Delta T$ before being released and the pump-probe scheme initiated. For each laser detuning, this process is repeated 100 times, with $\Delta T$ randomly sampled at each iteration to allow probing different statistical realizations of the complex spatio-temporal dynamics. Long enough loading periods are used to ensure acquisition occurs at steady-state conditions.

The lowest order statistical properties of atom density fluctuations can be described by the spatial auto-correlation function

$$c(\mathbf{r}_1, \mathbf{r}_2) = \langle \delta n(\mathbf{r}_1) \delta n(\mathbf{r}_2) \rangle, \qquad (1)$$

where $\delta n(\mathbf{r})$ is the local fluctuation density and the average operation is intended over the ensemble[18,19]. In the case of isotropic dynamics $c(\mathbf{r}_1, \mathbf{r}_2) \equiv c(r)$, the latter usually known as radial auto-correlation and related with the power spectrum of density fluctuations via Fourier transformation. Strong spatial correlations will be intrinsically related with the nucleation of photon bubble instabilities and the resulting regime of PBT.

**Turbulent regime characterisation**. The experimental results are summarized in Fig. 2. Panel a depicts the total power of the atom density fluctuations, obtained by integrating the power spectrum over all accessible wavenumbers, together with the corresponding correlation length. At ~$\delta = -2\Gamma$, the amplitude of density fluctuations sharply increases, coinciding with a peak in the correlation length. These features are indicative of a dynamical regime shift and the onset of a unstable phase. Panel b shows the full power spectrum in both stable and turbulent regions, with the observation of a sharp increase in the amplitude of density fluctuations in the entire range of accessible wavenumbers. This points to an exchange of energy between different length scales in a regime of fully developed turbulence. Panel c depicts the density auto-correlation function in the stable, transition and turbulent regions. The latter is characterized by a distinct correlation structure, suggesting the presence of large-amplitude quasi-coherent structures originating from photon bubbles.

It becomes pertinent at this point to discuss the physical origin of the turbulent regime illustrated above. The combined effects of a large number of atoms and near-resonant driving result in optically thick atom clouds. Light from the trapping and cooling beams is scattered multiple times, thus becoming diffusive. In regions of higher atom density, this can lead to significant photon trapping and a local increase of radiation-pressure forces. The atoms are slowly expelled, creating regions of high photon density and depleted atom presence, akin to photon bubbles. These, however, are short-lived and soon decay, leading to a re-establishment of the local atom density, which re-triggers the instability. This complex dynamics of continuous bubble growth and decay is at the root of the turbulent behaviour described here.

**Photon bubble instability**. The above arguments can be made more precise by developing a model describing the coupled dynamics of both photon and atom gases. In conditions of high optical thickness, photon transport can be described by the diffusion equation[20]

$$\frac{\partial}{\partial t} I - \nabla \cdot (D\nabla I) = 0, \qquad (2)$$

where $I$ is the local photon density. The diffusion coefficient is given by $D = l^2/\tau$, with $l$ the photon mean free path, inversely proportional to the atom density, and $\tau$ the photon diffusion time, usually considered independent from the atom density[21].

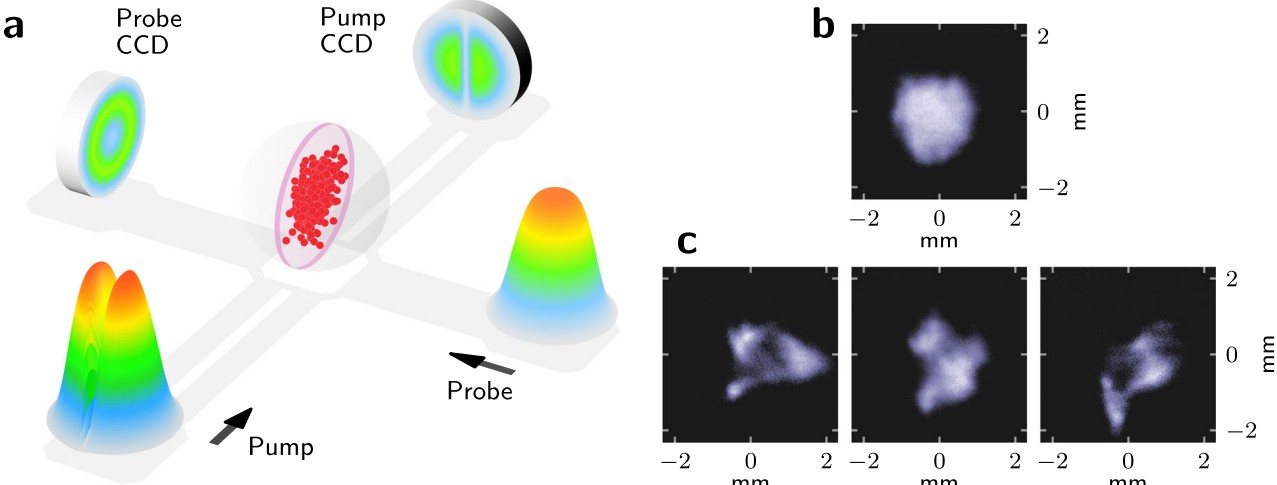

**Fig. 1 Direct imaging scheme and unfolding of the turbulent regime. a** Concept sketch illustrating the imaging working principle. The transversely carved pump beam induces a ground-state change in the atoms located in the outer shells of the cloud. A secondary beam propagating in an orthogonal direction then probes the remaining atoms located in a quasi-2D slab at the centre of the cloud. Typical atom density distributions in both the stable (**b**) and turbulent (**c**) regimes.

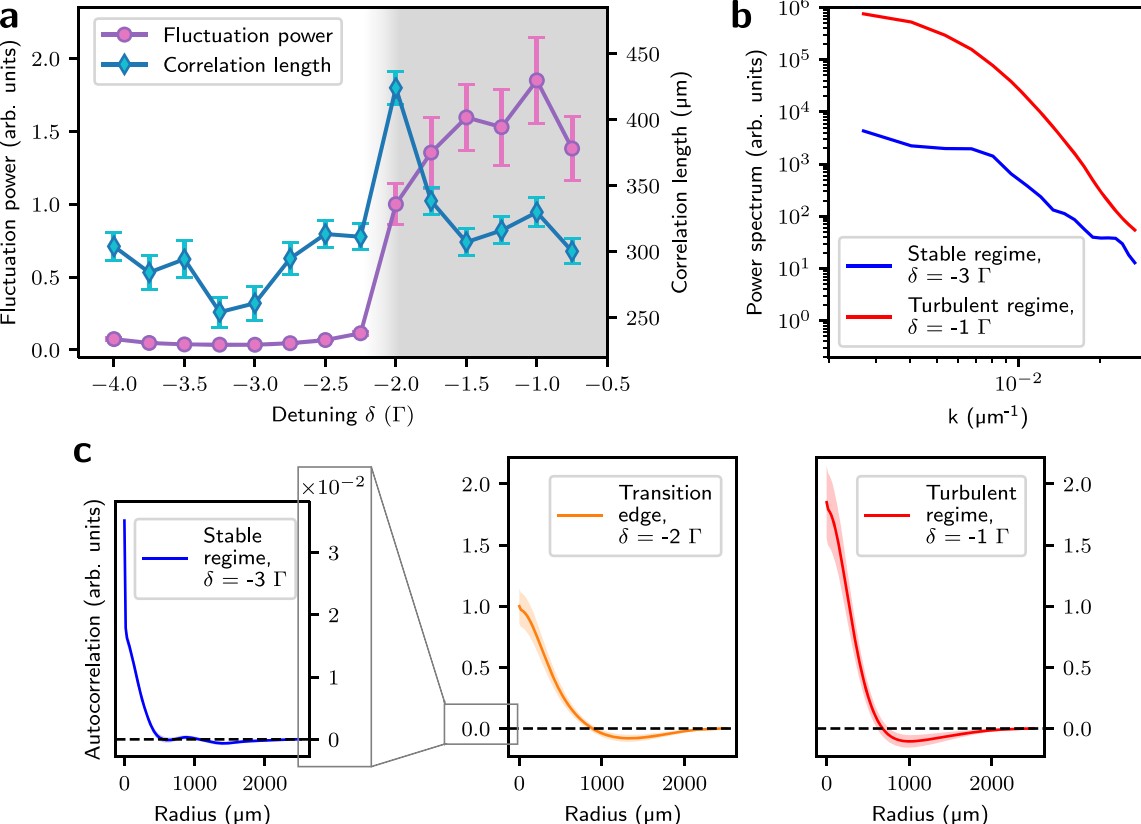

**Fig. 2 Transition into the regime of photon bubble turbulence. a** Correlation length and total power of relative atom density fluctuations as a function of laser detuning $\delta$, showing the transition into a turbulent regime where fluctuations are of the same order as the average density. Error bars correspond to $2\sigma$ variations over different realizations of the experiment. **b** Stable-turbulent transition in the power spectrum domain. **c** Typical radial auto-correlation function of density fluctuations in the stable, transition and turbulent regions.

Therefore, $D \propto n^{-2}$, which encodes the fact that regions of higher atom density give rise to slower photon diffusion. The resulting radiation-pressure forces are described by the Poisson equation[22,23]

$$\nabla \cdot \mathbf{F} = Qn. \tag{3}$$

The effective charge $Q$, which quantifies atom–atom repulsion mediated by photon exchange, is given by $Q = (\sigma_R - \sigma_L)\sigma_L I/c$, with $\sigma_L$ and $\sigma_R$ the photon absorption and re-scattering cross sections, respectively[23,24]. So, we have $Q \propto I$, with radiation-pressure forces being proportional to the local photon density $I$. The diffusion coefficient and the effective charge thus provide

the two-way coupling between light and matter that lies behind the nucleation of photon bubble instabilities[20].

The diffusion and Poisson equations above can be closed with the continuity and Navier–Stokes equations, namely[16,20,25]

$$\frac{\partial n}{\partial t} + \nabla \cdot (n\mathbf{v}) = 0, \qquad \text{and} \qquad (4)$$

$$\frac{\partial \mathbf{v}}{\partial t} + (\mathbf{v} \cdot \nabla)\mathbf{v} = \frac{\mathbf{F}}{m} - \frac{\nabla P}{nm} - \nu \mathbf{v}, \qquad (5)$$

where $m$, $P$ and $\nu$ are the atom mass, the thermodynamic pressure and the damping coefficient of the optical molasses, respectively. In order to investigate the stability of the system, we we will now be interested in the dynamics of small perturbations on top of equilibrium configurations of the form $n = n_0 + \tilde{n}$ and $I = I_0 + \tilde{I}$, with both $\tilde{n}$ and $\tilde{I}$ assumed to be of small amplitude. We begin by assuming, for the sake of simplicity, that the system is infinite and homogeneous. Later, we will develop a more robust model, which includes finite-size effects to better describe the experimental observations.

## Discussion

Under the above approximations, one can investigate the stability of the set of Eqs. (2)–(5), which has been carefully described elsewhere[20]. The analysis reveals the presence of unstable solutions triggered by inhomogeneities of the photon density of the form $\nabla^2 I_0/I_0 < 0$, corresponding to local regions of strong radiation trapping. Also, at the linear level, we find $\tilde{n} \propto -\tilde{I}$, meaning that a local increase of the photon density is accompanied by atom depletion due to radiation pressure, corresponding to the growth of a photon bubble. As the amplitude of these structures increases, nonlinear effects take place, leading to the saturation and eventual decay and bursting of these bubbles. The instability is thus continuously re-triggered, leading to complex spatio-temporal dynamics. Despite the complex turbulent behaviour, the phenomenology described here is intrinsically different from the one leading to the classical Kolmogorov-like turbulence. The cold atom gas is sufficiently diluted for inertia and viscosity not to play a significant role, thus preventing the transfer of kinetic energy from large to smaller scales through an energy cascade of the Kolmogorov, or other, kinds.

To further enquire about the properties of the turbulent regime, we shall now look for dynamical solutions to the line-arised form of the model equations. These can be generally written as $(\tilde{I}, \tilde{n}) \propto C(\mathbf{r})e^{-i\Omega t}$. These modes can become unstable, describing the depletion of the atom medium and the simultaneous growth of high photon density regions—photon bubbles. Here, $\Omega = \omega_r - i\gamma_i$ is a complex frequency, where $\omega_r$ is the oscillation frequency of the perturbed modes and $\gamma_i$ describes the amplitude growth of these unstable solutions. Going back to the fluid and diffusion equations, we observe that $C(\mathbf{r})$ satisfies a generalised Helmholtz equation

$$\nabla^2 C(\mathbf{r}) = -k_a^2 C(\mathbf{r}), \qquad (6)$$

where $k_a^2 = (q - i\gamma)^2$. Here, $q$ describes the spatial extent of the bubbles whereas $\gamma$ accounts for photon losses. Solutions to Eq. (6) can be conveniently expressed as

$$C(\mathbf{r}) = j_\ell(qr)Y_{\ell m}(\theta, \phi)e^{-\gamma r}, \qquad (7)$$

where $j_\ell(qr)$ are the spherical Bessel functions of order $\ell$ and $Y_{\ell m}$ are spherical harmonics.

Figure 3 depicts the full-2D as well as the radial-only experimental auto-correlation function, both at the transition region and deep in the turbulent regime. Despite the general absence of symmetry in the atom cloud during the turbulent regime, the auto-correlation function is essentially spherically symmetric.

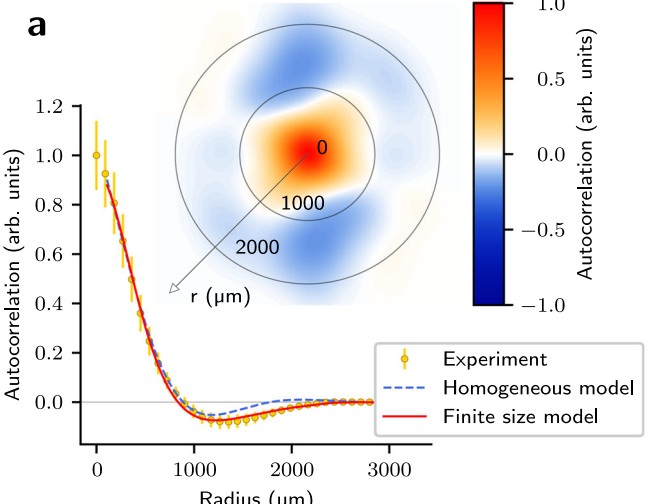

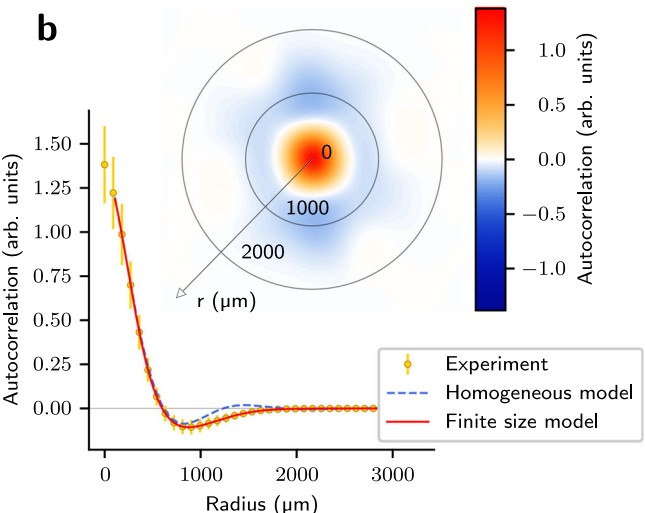

**Fig. 3 Comparison between experiment and photon bubble model.** Auto-correlation structure at the transition region, $\delta = -2\Gamma$ (**a**), and in the turbulent regime, $\delta = -0.75\Gamma$ (**b**). The blue dashed curve depicts the fit to the spherically symmetric solutions of the homogeneous photon bubble model. The red curve shows the improved fit when finite-size effects are taken into consideration. Both fits were performed only at scales larger than the probed layer thickness, where line-of-sight integration effects become negligible. Error bars have been defined in Fig. 2.

This is a result of isotropy of the photon scattering processes. Therefore, while at any given point in time no particular symmetry exists in the system, the statistics of atom density fluctuations are observed to maintain spherical symmetry.

The radial-only auto-correlation function can then be compared to the spherically symmetric solutions of the homogeneous model. By allowing $q$, related with the bubble size, and $\gamma$, the friction coefficient, as free fitting parameters, we obtain a relatively good agreement between theory and experiment. The observations are thus consistent with the picture of dynamical quasi-coherent bubble-like structures.

The homogeneous model, however, slightly deviates from the experiment at larger length scales. We have performed simulations correcting for finite-size effects, thus including the damping of bubbles at larger scales. With these modifications, further discussed in the Supplementary Information, we obtain a model that fully describes the experimental observations. In particular,

the typical photon bubble length scale corresponds to ~$1/q \simeq 200$ µm and we consistently observe, in the whole turbulent region, $\gamma \lesssim q$, which implies that diffusion losses happen on a larger scale than the bubble size, allowing bubbles to develop and grow.

We have thus described the experimental observation of PBT in optically thick clouds of cold atoms. In particular, we have shown via direct measures of the atom density fluctuations, the presence of dynamical quasi-coherent structures, related to the growth and collapse of photon bubbles. This interpretation is further supported by the excellent agreement with a theoretical model describing the coupled dynamics of both photon and atom gases. The conditions under which photon bubble instability occurs—randomised photon propagation, strong radiation-pressure forces—are analogous to those of dense astrophysical objects. In cold atoms, these are not the result of high densities but rather of the sharp electronic transitions and low temperatures, so that thermal and radiation-pressure effects become comparable. Our research shows that cold atom experiments can be exploited to investigate instabilities similar to those occurring in complex space plasmas. However, the complete understanding of the relevant differences and similarities between the two systems is still under debate and will require further examination.

## Methods

**Magneto-optical trap (MOT).** Laser cooling and trapping is performed on the $F = 3 \rightarrow F' = 4$ transition of the 2D line of Rubidium 85, at ~780 nm. The lasers are slightly red-detuned from exact resonance by $\delta$. The six independent beams cross the centre of the trap with a beam waist of about 4 cm and power per beam $P \sim 40$ mW. Atoms are collected from a dilute vapour at a background pressure of ~$10^{-8}$ Torr. Due to the finite probability of exciting the open $F = 3 \rightarrow F' = 3$ transition, some atoms undergo a ground-state change via a spontaneous Raman transition. The cooling and trapping transition is recycled by a repump beam tuned to the $F = 2 \rightarrow F' = 3$ transition. Trapping by radiation pressure is achieved through a magnetic field gradient of ~10 G/cm created by a pair of coils in an anti-Helmholtz configuration[16,17], inducing a Zeeman split of the electronic states. A full scheme of the experiment, including details on the pump and probe imaging diagnostics can be found in the Supplementary Information.

**Data treatment and analysis.** Upon propagation through the thin quasi-2D atom slab, the probe beam gets partially absorbed and its output intensity—$I_A(x, y)$—is imaged onto a CCD. For each realization, two more images are collected: an image of the probe beam without the atom cloud—$I_{NA}(x, y)$—and a dark background image with no laser—$I_D(x, y)$. Using the Lambert–Beer law, it is possible to retrieve the background-corrected local optical density of the atom layer by

$$od(x, y) = \log\left(\frac{I_{NA}(x, y) - I_D(x, y)}{I_A(x, y) - I_D(x, y)}\right). \tag{8}$$

From the optical density we can directly map the density of the thin quasi-2D atom slab—$n_i(x, y)$, with $i$ labelling the different realizations of the experiment.

We define $\langle n(x, y)\rangle$ as atom density averaged over all the $i = 1, 2,...,100$ realisations of a given experiment. The density fluctuation for each $i$th realization are then given by $\delta n_i(x, y) = n_i(x, y) - \alpha_i\langle n(x, y)\rangle$. Here, the coefficient $\alpha_i$ is calculated as to filter out effects coming from variations of the total number of atoms at different realizations. It is defined as

$$\alpha_i = \frac{\sum\limits_{x,y} n_i(x, y)}{\sum\limits_{x,y} \langle n(x, y)\rangle}, \tag{9}$$

where $\sum_{x,y}$ is meant as summation over all camera pixels.

To examine the statistical nature of atom density fluctuations, we calculate the full-2D auto-correlation function for each $\delta n_i(x, y)$, which are then averaged over the different realizations. The isotropic contribution is obtained by averaging over all the polar coordinate, which finally gives the radial auto-correlation function $c(r)$. The corresponding correlation length is defined as

$$L = \frac{\sum\limits_{k} |g(r_k) r_k|}{\sum\limits_{k} |g(r_k)|}. \tag{10}$$

## Data availability

All the data supporting this work are available from the corresponding author upon reasonable request.

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

## Acknowledgements

We thank Hugo Terças for stimulating discussions on turbulence and the general physics of cold atom gases. We thank R. Kaiser for helpful discussions during the initial stages of the MOT experiment. This work has received funding from the European Union's Horizon 2020 Research and Innovation programme under grant agreement no. 820392 (PhoQuS). R.G. acknowledges the Advanced Programme in Plasma Science and Engineering (APPLAuSE) and the financial support of FCT (Fundação para a Ciência e Tecnologia) through the Grant Number PD/BD/135211/2017.

## Author contributions

R.G. designed and implemented the diagnostic, performed the experiments, analysed the data and wrote the bulk of the manuscript with contributions from all the authors. J.D.R. co-supervised the project, conceived the experiment, contributed to analysing the data and interpreting the results. J.A.R. contributed to assembling and developing the experimental setup. J.T.M. supervised the project. All authors contributed to the scientific discussion of the results.

## Competing interests

The authors declare no competing interests.
