## [Peer Review File · Nature Communications]

REVIEWER COMMENTS

Reviewer #1 (Remarks to the Author):

In this work the authors report on the observation of photon bubble turbulence in a MOT of rubidium atoms. They access the turbulent regime by exciting the atoms close to the absorption transition: the turbulent regime is demonstrated by measuring in a time-resolved fashion the atom entity distribution after the excitation. They model the physics of the system both for photons than for the atomic sample.

I find this work interesting, and well written.

I could recommend the publication in Nature Communications, provided that the authors clarify some points in the text:

1) The first question regards the definition of turbulent regime. In general context, the turbulent regime is associated to the Kolmogorov cascade, which describe the behavior of the momentum distribution. How this concept is applicable here? Which is the Kolmogorov scale in this experiment?

2) Since the excitation of the cloud close happens using almost resonant light, which is the role of heating and of dissipation in the observed dynamics?

3) In turbulence, vortex tangling (in some sense, vortex-vortex interactions) plays a decisive role. Which is, if present, the role of bubble-bubble interactions?

4) Is there any similar phenomenology if the excitation is placed at +2 Gamma instead of being at -2 Gamma (with respect to the optical transition)? May we expect a symmetric behavior across the resonance?

5) Typical densities in the MOT are of the order of few 10^9 at/cm³. What would happen in the case of denser cloud, eventually trapped in an optical dipole trap?

6) In the model presented, I do not fully get why the turbulent regime should be described by a dynamical solution of the form $(I,n) \sim C(r) \exp^{-(i \Omega)t}$. May the authors comment more about? What is Omega?

7) I would like the authors to clarify and in case reinforce the final sentences: "The conditions under which photon bubble instabilities occur in cold atoms are analogous to those of dense astrophysical objects. As a result, we have laid the groundwork for future investigations of space plasma phenomena in laboratory based experiments, therefore creating a viable testbed for complex astrophysical processes": which are indeed the next steps for future investigations, to link even more these two scenarios? I found these conclusions weak and not completely defined.

Thanks

Reviewer #2 (Remarks to the Author):

This paper, Photon Bubble Turbulence... by Giampaoli et al., describes and interprets an experiment involving a disk-like arrangement of cold Rb atoms in a magneto-optical trap. The paper shows the development of an unsteady state whose onset is triggered by tuning the laser toward resonance. The paper analyzes the unsteady state in terms of a linear stability analysis by Mendonca and Kaiser, and argues that the unsteady

state is analogous to the photon bubble instability in astrophysics.

The paper is interesting, potentially important, and potentially appropriate for publication in Nature, but the authors need to address the issues raised below.

major comments:

This paper presents results similar to <https://arxiv.org/pdf/1604.08114.pdf>
How does it differ?

The interpretation in terms of the Mendonca-Kaiser instability is given without a persuasive discussion that (1) the approximations used in the Mendonca-Kaiser analysis are relevant here, and (2) that the Mendonca Kaiser instability is relevant to astrophysical systems.

minor comments:

What is the optical depth of the atoms in the trap? This is determines the applicability of the diffusion approximation (eq. 2).

The structure in fig 1c looks anything but spherically symmetric. Why should the spherically symmetric solution be relevant? It is evident that whatever instabilities are present break spherical symmetry.

The statement that the atomic density fluctuations are fully described by the auto-correlation functions is not strictly true. Higher order correlations may also be needed (as in, e.g., turbulence). For a gaussian random field the autocorrelation function provides a complete description.

Reviewer #3 (Remarks to the Author):

This is basically a study of atomic spatial density distribution as well as its fluctuation autocorrelation function inside a MOT. Experimentally, a clever pump-probe scheme is developed, which allows for imaging the density of atoms in a spatial slice of a MOT with a near resonant probe laser, after atoms in the outer regions are pumped away by a near resonant laser (with a "rod" caused shadow which keeps atoms inside the imaged slice from being pumped), all carried out in a brief time window after lasers forming the MOT are turned off. The authors interpret the observed density autocorrelation, particularly its spectrum density distribution, whose dependence on the MOT laser detuning to reveal characteristically different atomic cloud distributions, from relatively smooth varying ones in space to turbulent with coherent patterns indicative of cavitation associated with photon-bubbles.

The paper is well written, and the story is well told. But I don't feel the corroboration between theory and experiments are sufficiently strong that the photon-bubble story is solidly established. It is long known that atomic spatial distribution inside a MOT is complicated and inhomogeneous with large fluctuations. When optical density is high, propagation and transmission of photons scattered by atoms from MOT forming lasers turns into diffusive, accompanied by losses of all types, from radiative transfer to collisional redistribution of light in the far wings of the spectrum when absorption and rescattering by an atom occurs under the influence of other nearby atoms, etc. But in the supplemental material, the measured optical density for the pump laser light is actually not all that high, how does this translate into the original MOT? which is a typical 3D type.

In summary, I am not convinced of the presented story, certainly, not by the presented corroborations between experimental observations and theoretical modelling or data fitting.

Answers to Reviewer 1

We appreciate the scrupulous and diligent review; the remarks raised by the referee have been beneficial to convey a clearer and more coherent message throughout the paper. We have addressed the referee's response point by point below, quoting the modifications to the manuscript.

Comment 1: *The first question regards the definition of turbulent regime. In general context, the turbulent regime is associated to the Kolmogorov cascade, which describe the behavior of the momentum distribution. How this concept is applicable here? Which is the Kolmogorov scale in this experiment?*

REPLY: The classical concept of Kolmogorov turbulence is not applicable here. The photon bubble dynamics is driven by fundamentally different processes. The Kolmogorov cascade results from the presence of vorticity eddies that carry the energy that is injected into the larger scales onto the much smaller scales where dissipation by viscosity occurs. As a result of inertia, these vorticity eddies develop at a continuum of length scales (the inertial range) where the momentum distribution typically shows the universal Kolmogorov power spectrum. In our system, the atom gas is sufficiently diluted such that inertia and viscosity do not play a significant role. Instead, the dynamics are dominated by radiation pressure forces coupled to the diffusive nature of light in optically thick regions of the atom cloud. The turbulent behavior emerges as a consequence of the complex and continuous cycles of bubble growth, resulting in local atom depletion due to radiation pressure, and bubble burst, because of the decreased optical density.

In order to address this important distinction, we are including the following sentence in the manuscript, right after the paragraph in which the model is introduced:

Despite the complex turbulent behavior, the phenomenology described here is intrinsically different from the one leading to the classical Kolmogorov-like turbulence. The cold atom gas is sufficiently diluted for inertia and viscosity not to play a significant role, thus preventing the transfer of kinetic energy from large to smaller scales through an energy cascade of the Kolmogorov, or other, kinds.

Comment 2: *Since the excitation of the cloud close happens using almost resonant light, which is the role of heating and of dissipation in the observed dynamics?*

REPLY: The turbulent dynamics arise spontaneously as the trap is continuously maintained by the slight red-detuned laser beams, thus maintaining laser cooling and trapping while the photon bubble instabilities develop. The entire process, including cooling and trapping, is dissipative. In order to make the experimental procedure more clear, we have modified the following:

~~A transition from a stable to a turbulent regime is observed when the frequency of the cooling and trapping beams is brought close to that of the electronic transition.~~

The trap is continuously maintained by the cooling and trapping beams, with a transition from a stable to a turbulent regime being observed as we bring the frequency of these lasers close to that of the electronic transition.

Comment 3: *In turbulence, vortex tangling (in some sense, vortex-vortex interactions) plays a decisive role. Which is, if present, the role of bubble-bubble interactions?*

REPLY: Bubble-bubble interactions are not expected to play a significant role here for the following reasons:

1) The typical bubble size is only slightly smaller than the total atom cloud size, meaning that,

on average, the existence of multiple regions of developed bubble growth is highly suppressed;
 2) Bubble-bubble interactions, if present, would be relevant at the non-linear stages of the instability. As shown in the main text, with further support shown in the supplemental material, in the limits of our experimental resolution, all the experimental observations are accurately described by our linear theoretical model, meaning that nonlinear effects associated with bubble-bubble interactions are not dominant, at least in the scales accessible and described here.

Comment 4: *Is there any similar phenomenology if the excitation is placed at +2 Gamma instead of being at -2 Gamma (with respect to the optical transition)? May we expect a symmetric behavior across the resonance?*

REPLY: The experiment cannot be performed in the blue side. The instability develops while laser cooling and trapping is continuously maintained, which can only occur in the red-side of the electronic transition. We believe the changes made as a reply to Comment 2 elucidate this point as well.

Comment 5: *Typical densities in the MOT are of the order of few 10^9 at/cm³. What would happen in the case of denser cloud, eventually trapped in an optical dipole trap?*

REPLY: While MOT densities are limited by multiple scattering of light to about 10^9 atoms/cm³, a larger number of atoms can be trapped by more powerful lasers. This would result in a larger optical thickness and a higher aspect ratio between bubble and system sizes, eventually proving access to the effects resulting from bubble bubble interactions and/or the development of turbulence over wider length scales. Regarding optical dipoles traps, these are not appropriate to conduct these experiments because cooling cannot be continuously maintained.

Comment 6: *In the model presented, I do not fully get why the turbulent regime should be described by a dynamical solution of the form $(\tilde{I}, \tilde{n}) \propto C(\mathbf{r})e^{-i\Omega t}$. May the authors comment more about? What is Omega?*

REPLY: As stated in the main text, from the linearised form of the model equations, we can show that $\tilde{I} \propto -\tilde{n}$. Thus, \tilde{I} and \tilde{n} share solutions of the same form. At the linear level, we can generally assume a separable form of these solutions as $(\tilde{I}, \tilde{n}) \propto C(\mathbf{r})e^{-i\Omega t}$, where $\Omega = \omega_r - i\gamma_i$ is a complex frequency with ω_r the oscillation frequency and γ_i describing the amplitude growth of these unstable solutions. By plugging this ansatz into the linearised fluid equations, $C(\mathbf{r})$ is shown to obey an Helmholtz equation, whose solution can be expanded in a basis of Bessel functions, as we explain further in the text. This procedure is completely generic. Also, as we show in the manuscript, the experimental correlation functions are essentially spherically symmetric, so we restrict our theoretical analysis to spherically symmetric contributions to $C(\mathbf{r})$. We have briefly modified the manuscript to further describe these details:

~~To further inquire about the properties of the turbulent regime, we shall now look for dynamical solutions of the form $(\tilde{I}, \tilde{n}) \propto C(\mathbf{r})e^{-i\Omega t}$. Going back to the fluid and diffusion equations, we observe that $C(\mathbf{r})$ satisfies a generalised Helmholtz equation $\nabla^2 C(\mathbf{r}) = -k_a^2 C(\mathbf{r})$ where $k_a^2 = (q - i\gamma)^2$. Here, q describes the spatial extent of the bubbles whereas γ accounts for photon losses.~~

To further inquire about the properties of the turbulent regime, we shall now look for dynamical solutions to the linearised form of the model equations. These can be generally written as $(\tilde{I}, \tilde{n}) \propto C(\mathbf{r})e^{-i\Omega t}$. These modes can become unstable, describing the depletion of the atom medium and the simultaneous growth of high photon density regions – photon bubbles. Here, $\Omega = \omega_r - i\gamma_i$ is a

complex frequency where ω_r is the oscillation frequency of the perturbed modes and γ_i describes the amplitude growth of these unstable solutions. Going back to the fluid and diffusion equations, we observe that $C(\mathbf{r})$ satisfies a generalised Helmholtz equation.

$$\nabla^2 C(\mathbf{r}) = -k_a^2 C(\mathbf{r}),$$

where $k_a^2 = (q - i\gamma)^2$. Here, q describes the spatial extent of the bubbles whereas γ accounts for photon losses.

Comment 7: *I would like the authors to clarify and in case reinforce the final sentences: "The conditions under which photon bubble instabilities occur in cold atoms are analogous to those of dense astrophysical objects. As a result, we have laid the groundwork for future investigations of space plasma phenomena in laboratory based experiments, therefore creating a viable testbed for complex astrophysical processes": which are indeed the next steps for future investigations, to link even more these two scenarios? I found these conclusions weak and not completely defined.*

REPLY: We are including the following modification to the manuscript as a reply:

~~The conditions under which photon bubble instabilities occur in cold atoms are analogous to those of dense astrophysical objects. As a result, we have laid the groundwork for future investigations of space plasma phenomena in laboratory based experiments, therefore creating a viable testbed for complex astrophysical processes~~

The conditions under which photon bubble instability occurs – randomised photon propagation, strong radiation-pressure forces – are analogous to those of dense astrophysical objects. In cold atoms, these are not the result of high densities but rather of the sharp electronic transitions and low temperatures, so that thermal and radiation pressure effects become comparable. By experimentally demonstrating a close connection between such two disparate physical systems, we have shown the possibility to use cold atoms as a viable testbed to investigate complex space plasma processes.

Answers to Reviewer 2

We thank the reviewer for providing accurate and significant comments. The remarks have been certainly useful to make the paper more sound and precise. We have addressed the referee's response point by point below, quoting the modifications to the manuscript.

Comment 1: *This paper presents results similar to <https://arxiv.org/pdf/1604.08114.pdf>. How does it differ?*

REPLY: The paper in arxiv is a paper we submitted few years ago on the same topic that never got published. It was based on a different set of data gathered by collecting the light scattered from the entire atom cloud. The spatial structures resulting from the nucleation of the instability were then analysed assuming the whole atom cloud to be spherically symmetric. A major critic about the previous discussion was related with the diagnostics then available: scattering measures rely on a line-of-sight integration signal which implies not having direct access to the local atom density unless assuming, a priori, a specific symmetry of the atom cloud.

To overcome this limitation, we began developing an improved diagnostics technique, the one described in this submission. This new diagnostic allowed to directly access the local atom density, with no assumption about the shape of the cloud. We thus re-examined the subject with more significant and compelling experimental data involving direct measurements of the local atom density. The present conclusions are consistent with that previous unpublished work.

Comment 2: *The interpretation in terms of the Mendonça-Kaiser instability is given without a persuasive discussion that:*

- (1) *the approximations used in the Mendonça-Kaiser analysis are relevant here,*
- (2) *the Mendonça-Kaiser instability is relevant to astrophysical systems.*

REPLY:

(1) The fundamental approximation used in the instability model is the diffusion approximation, which will be addressed in the reply to comment 3.

(2) Strictly speaking, a similar model of photon bubble instability has been proposed as a mechanism responsible for the formation of proto-planets and voids in self-gravitating dust gases such the interstellar medium or in galactic loops (Photon Bubbles in a Self-gravitating Dust Gas: Collective Dust Interactions, <https://iopscience.iop.org/article/10.3847/1538-4357/aafe7e/meta>). Sure enough, the conditions at which instabilities arise in the Mendonça-Kaiser model differ from the ones in hot dense plasmas (Photon bubbles - Overstability in a magnetized atmosphere, <http://adsabs.harvard.edu/full/1992ApJ...388..561A>): most notably, the instability arising in cold atom media does not require a magnetic field. However, both in cold atom and massive astrophysical objects, the system is described by a set of coupled fluid equations for both the photon and matter fluids and the instability results from the pressure exerted on the matter fluid by diffusing radiation. The resulting turbulent regime in cold atoms is akin to the one in hot plasma as they exhibit analogous dynamics. In both media, the continuous process of radiation-filled pockets growing and bursting defines a pattern of high atom density region alternating with regions almost devoid of particles. Hence, the two phenomena are closely related so that deepen our knowledge of the cold atom system would provide crucial information on the unfolding of its high-temperature counterpart. As mentioned in the introduction, this has considerable implications in the understanding of the enhanced radiation and energy transport in space plasmas.

In order to further explicit the link between cold atom and astrophysical dynamics, we modified

the conclusions to the paper as follows:

~~The conditions under which photon bubble instabilities occur in cold atoms are analogous to those of dense astrophysical objects. As a result, we have laid the groundwork for future investigations of space plasma phenomena in laboratory-based experiments, therefore creating a viable testbed for complex astrophysical processes~~

The conditions under which photon bubble instability occurs – randomised photon propagation, strong radiation-pressure forces – are analogous to those of dense astrophysical objects. In cold atoms, these are not the result of high densities but rather of the sharp electronic transitions and low temperatures, so that thermal and radiation pressure effects become comparable. By experimentally demonstrating a close connection between such two disparate physical systems, we have shown the possibility to use cold atoms as a viable testbed to investigate complex space plasma processes.

Comment 3: *What is the optical depth of the atoms in the trap? This determines the applicability of the diffusion approximation (eq. 2).*

REPLY: Being a critical point to the entire paper, we have prepared an entire new section in the Supplementary Information - Section 2, 'Diffusion approximation' - where we describe experimental measurements of optical density and, based on them, conduct simple Monte-Carlo simulations elucidating on the statistics of the photon scattering processes and, hence, to which extent the diffusion approximation holds. We invite the Referee to analyse this newly added section.

Comment 4: *The structure in fig 1c looks anything but spherically symmetric. Why should the spherically symmetric solution be relevant? It is evident that whatever instabilities are present break spherical symmetry.*

REPLY: Indeed, Fig 1.c clearly shows that the atom cloud in the unstable regime is not spherically symmetric. However, we do not make any assumption about the shape of the atom cloud. Spherical symmetry is observed not on the density distribution, but rather on the statistical properties of the density perturbations. In Fig. 3, without assuming any kind of symmetry in the system, we show that the 2D auto-correlation function is essentially spherically symmetric, which follow from the isotropy of the processes taking place in the atom cloud. The instability, however, can be nucleated in different positions of the atomic cloud and therefore the cloud shape at any given moment has no particular symmetry.

We make this point more explicit with the following modification to the manuscript:

~~Given that all relevant processes are essentially isotropic, the auto-correlation structure of the photon bubble regime should be dominated by spherically symmetric solutions $\ell = 0$. The comparison between experiment and the spherically symmetric solutions of the homogeneous model is depicted in Fig.3, both at the transition region and deep in the turbulent regime.~~

Fig.3 depicts the full-2D as well as the radial-only experimental auto-correlation function, both at the transition region and deep in the turbulent regime. Despite the general absence of symmetry in the atom cloud during the turbulent regime, the auto-correlation function is essentially spherically symmetric. This is a result of isotropy of the photon scattering processes. Therefore, while at any given point in time no particular symmetry exists in the system, the statistics of atom density fluctuations are observed to maintain spherical symmetry.

The radial-only auto-correlation function can then be compared to the spherically symmetric solutions of the homogeneous model.

Comment 5: *The statement that the atom density fluctuations are fully described by the auto-correlation functions is not strictly true. Higher order correlations may also be needed (as in, e.g., turbulence). For a gaussian random field the autocorrelation function provides a complete description.*

REPLY: We agree with the Referee. We have modified the manuscript accordingly:

~~The statistical properties of atom density fluctuations are fully described by the spatial auto-correlation function~~

The lowest order statistical properties of atom density fluctuations can be described by the spatial auto-correlation function

Answers to Reviewer 3

Comment 1: *The paper is well written, and the story is well told. But I don't feel the corroboration between theory and experiments are sufficiently strong that the photon-bubble story is solidly established. It is long known that atom spatial distribution inside a MOT is complicated and inhomogeneous with large fluctuations. When optical density is high, propagation and transmission of photons scattered by atoms from MOT forming lasers turns into diffusive, accompanied by losses of all types, from radiative transfer to collisional redistribution of light in the far wings of the spectrum when absorption and rescattering by an atom occurs under the influence of other nearby atoms, etc. But in the supplemental material, the measured optical density for the pump laser light is actually not all that high, how does this translate into the original MOT? which is a typical 3D type.*

REPLY: We acknowledge that a proper justification of the diffusion approximation is critical to the message we want to convey here. As such, we have prepared an entire new section in the Supplementary Information - Section 2, 'Diffusion approximation' - where we describe experimental measurements of optical density and, based on them, conduct simple Monte-Carlo simulations elucidating on the statistics of the photon scattering processes and, hence, to which extent the diffusion approximation holds. We invite the Referee to analyse this newly added section.

Reviewers' Comments:

Reviewer #1:

Remarks to the Author:

The authors have provided clear answers and explanations to all of my criticisms. In my opinion, now the paper meets the criteria for the publication in Nature Communications without any additional changes.

Regards

Reviewer #2:

Remarks to the Author:

This revised version improves significantly on the earlier version. The discussion of optical depth in the supplemental material is especially useful, although it suggests that the Rb atom cloud is only marginally optically thick.

The paper is very interesting, and the result merits publication.

The remaining issue is the claim that is being made:

"By experimentally demonstrating a close connection between such two disparate physical systems, we have shown the possibility to use cold atoms as a viable testbed to investigate complex space plasma processes."
[from the rebuttal document and revised paper]

This isn't entirely accurate. Rather the authors have exhibited the development of strong fluctuations in the atomic cloud and shown that - to the extent it can be tested - the fluctuations are consistent with turbulence driven by the photon bubble instability of Mendonca and Kaiser. This is intriguing because it opens the possibility of using cold atoms as a testbed to investigate complex space plasma processes.

The difference between the claims in the paper and what has been actually shown are important: (1) the observed fluctuations cannot be claimed to be fully understood based on the limited data available, and clearly merit further investigation; (2) the relationship between the Mendonca-Kaiser instability and the instabilities (there are several) that are called a photon bubble instability in an astrophysical context remains unclear; (3) the usefulness of this experiment as a testbed for those instabilities therefore has not been fully demonstrated.

Reviewer #3:

Remarks to the Author:

I appreciate the authors for including in the supplemental material a section on the diffusion approximation for the scattered light. My earlier question raised the issue of near resonance optical density for the probe light in the MOT cloud. Upon studying the newly included section, which outlines the approximations involved, the experimental data, and numerical simulations, I don't feel the basis for photon diffusion is sufficiently established, therefore I do not accept the idea of a photon bubbled based on such a crude model. More specific reasons are given in the following:

1), The observed optical densities are clearly not high enough (smaller than unity for most cases) to support treating scattered photons, as shown in Fig. S3. The definitions given in Eqs. (1) and (2) also ignores important dependencies on various parameters to be mentioned in the following two points.

2), Inside a MOT, atomic density distribution is inhomogeneous, and known to sensitively depend on the misalignment of pumping lasers forming the MOT; an illustrative example can be found on page 163 of the book "laser cooling and trapping" by Metcalf et. al. For the present photon bubble model to stand, it would be necessary that the authors explain the size of the bubble or make a prediction of it based on the physical quantities such as laser fields detuning, atomic density, atomic mass and temperature, spontaneous decay linewidth, MOT magnetic field gradient, and pumping field parameters.

3), The absorption line-shape for the scattered photons are taken to be a fixed Lorentzian, independent of the location, atomic velocity, magnetic field gradient, or the continued presence of the pumping fields for the MOT. This is clearly inconsistent with the assumption that the whole story of photo bubble develops in the background of laser cooling and trapping processes maintained inside the MOT. The Mollow triplet spectra for the scattered light, underlines quantum coherence and correlation properties of scattered photons are also thrown away, while coherent components of the probe and the pump fields are assumed to dominant the atom-photon interaction processes.

Answers to Reviewer 2

Comment 1: *The remaining issue is the claim that is being made: "By experimentally demonstrating a close connection between such two disparate physical systems, we have shown the possibility to use cold atoms as a viable testbed to investigate complex space plasma processes." [from the rebuttal document and revised paper]*

This isn't entirely accurate. Rather the authors have exhibited the development of strong fluctuations in the atomic cloud and shown that - to the extent it can be tested - the fluctuations are consistent with turbulence driven by the photon bubble instability of Mendonca and Kaiser. This is intriguing because it opens the possibility of using cold atoms as a testbed to investigate complex space plasma processes.

The difference between the claims in the paper and what has been actually shown are important: (1) the observed fluctuations cannot be claimed to be fully understood based on the limited data available, and clearly merit further investigation; (2) the relationship between the Mendonca-Kaiser instability and the instabilities (there are several) that are called a photon bubble instability in an astrophysical context remains unclear; (3) the usefulness of this experiment as a testbed for those instabilities therefore has not been fully demonstrated.

REPLY: We acknowledge the pertinence of the referee's comment. We have revised this concluding remark as:

~~By experimentally demonstrating a close connection between such two disparate physical systems, we have shown the possibility to use cold atoms as a viable testbed to investigate complex space plasma processes.~~

Our research shows that cold atom experiments can be exploited to investigate instabilities similar to those occurring in complex space plasmas. However, the complete understanding of the relevant differences and similarities between the two systems is still under debate and will require further examination.

Answers to Reviewer 3

Comment 1: *The observed optical densities are clearly not high enough (smaller than unity for most cases) to support treating scattered photons, as shown in Fig. S3. The definitions given in Eqs. (1) and (2) also ignores important dependencies on various parameters to be mentioned in the following two points.*

REPLY: Fig.S3 belongs to Section I of the Supplementary Information (SI). This figure concerns the calibration of the pump-probe diagnostic and it is thus not pertinent to discuss photon diffusion. The systematic analysis of the diffusion approximation has been described in the following section of the SI. Moreover, due to the reasons we state in Section II of the SI, the optical density is a very limited quantity to assess the nature of photon transport when the MOT is continuously maintained. For these reasons, photon transport within the MOT needs to be addressed differently, for instance, by simulations of the kind we describe in Section II. These allow us to recover the full photon scattering statistics, under some assumptions and simplifications necessary to make the problem tractable. The results of these simulations are summarised in Fig. S6. The limitations of this procedure are further discussed in the following points.

Comment 2: *Inside a MOT, atomic density distribution is inhomogeneous, and known to sensitively depend on the misalignment of pumping lasers forming the MOT; an illustrative example can be found on page 163 of the book "laser cooling and trapping" by Metcalf et. al. For the present photon bubble model to stand, it would be necessary that the authors explain the size of the bubble or make a prediction of it based on the physical quantities such as laser fields detuning, atomic density, atomic mass and temperature, spontaneous decay linewidth, MOT magnetic field gradient, and pumping field parameters.*

Comment 3: *The absorption line-shape for the scattered photons are taken to be a fixed Lorentzian, independent of the location, atomic velocity, magnetic field gradient, or the continued presence of the pumping fields for the MOT. This is clearly inconsistent with the assumption that the whole story of photo bubble develops in the background of laser cooling and trapping processes maintained inside the MOT. The Mollow triplet spectra for the scattered light, underlines quantum coherence and correlation properties of scattered photons are also thrown away, while coherent components of the probe and the pump fields are assumed to dominant the atom-photon interaction processes.*

REPLY: We reply here to comments 2 and 3 jointly.

Predicting the dependence of the bubble size on experimental parameters like detuning, ∇B , density, atomic mass, temperature, spontaneous decay linewidth, laser parameters, etc, is a very complicated problem which would ultimately involve not only the precise knowledge of the scattering cross sections and their dependencies on all these parameters, but, also, the knowledge of how these cross sections influence the full photon scattering statistics. Even the understanding of these cross sections alone has been, and remains, the subject of strong scientific debate. Check, for instance:

- Labeyrie *et al*, Physical review letters **96** (2), 023003 (2006)
- Mendonça *et al*, Physical Review A **78** (1), 013408 (2008)
- Gattobigio *et al*, Physica Scripta **81** (2), 025301 (2010)
- Camara *et al*, Physical Review A **90** (6), 063404 (2014)
- Ortiz-Gutiérrez *et al*, New Journal of Physics **21** (9) 093019 (2019)

- Barre *et al*, Physical Review A **100** (1), 013624 (2019)
- Gaudesius *et al*, Physical Review A **101** (5), 053626 (2020)

In this work, we took the following approach to shed some light on this problem:

- Many of the dependencies mentioned above are reflected onto the MOT optical density, which we are empirically approximating by measuring the optical density using the pump and probe experiment. This measurement, however, ignores dependencies such as magnetic field-induced spatial inhomogeneities and effects from the strong driving fields, including quantum coherence.
- Building on these measurements of the optical density, some of the intricate dynamics of photon scattering arising in the MOT are reintroduced into the Monte Carlo model that we describe in Section II, like spontaneous decay linewidth, detuning dependence and asymmetry of absorption and emission processes.
- Despite these inherent limitations, the Monte Carlo simulations approximate the full statistics of photon transport, which, as said before, cannot be directly inferred from models of cross sections alone.

The approximations above are not inconsistent with the picture of photon diffusion developing in the background of laser cooling and trapping. Despite ignoring spatial dependencies, atomic velocity, magnetic field gradient and the presence of the Mollow triplet, not only the most relevant cross section dependencies on the experimental parameters are built into the model via the pump and probe measurements of the optical density, but, also, the construction of the Monte Carlo simulations themselves reintroduces some of the processes specific to the MOT.

The results so obtained, despite the approximations undertaken, are in complete agreement with the picture of a diffusive contribution to the full photon dynamics (including cooling and trapping). This behavior is not persistent across the entire range of experimental parameters. Rather, the simulations show a strong and sharp transition into this diffusive regime in the same range of parameters where the turbulent regime is experimentally identified. This strongly corroborates the approximations inherent to our theoretical model and, hence, the origin of the observed turbulent dynamics in the photon bubble instability mechanism.

Going back to the question about estimating the bubble size, while a precise prediction is limited by all the reasons stated above, let us, in a much simplified picture, show that the results of the Monte Carlo simulations are consistent with the scales observed in the experiment. Let's restrict our attention to the turbulent regime. In particular, Fig.R1 shows the entire distribution of scattering events obtained from the Monte Carlo simulations. The corresponding mean number of scattering events, conditioning on a photon being scattered at least once, is $n_s \simeq 4.0$. Without taking into account geometrical factors, the effective optical density and the mean number of scattering events are related by $b^* = \sqrt{n_s}$ [Labeyrie *et al*, Physical review letters **91** (22), 223904 (2003)], with the effective optical density being related with the photon mean free path ℓ as $b^* = L/\ell$, with L the system size. Putting all these numbers together, considering a typical system size of about 4 mm, we obtain $\ell \sim 2$ mm, which is of the same order of the bubble size (twice the length scale observed in the radial auto-correlation function). This estimation is highly hand-wavy, as it ignores not only all the dependencies described above but also the full statistics of photon scattering. In particular, we are not distinguishing here between the single-scattered photons that maintain cooling and trapping, and the smaller diffusive background responsible for the nucleation of the instability. In any case, we can show that the scales involved are consistent with our picture.

Fig. R1: **Photon scattering event distribution.** The figure shows the distribution of scattering events in the turbulent regime ($\delta_{\text{MOT}} = -1\Gamma$), obtained from the Monte Carlo simulations described in Section II of the SI. The photons that are initially absorbed by the atomic cloud scatter, on average, 4 times.

Reviewers' Comments:

Reviewer #2:

None

Reviewer #3:

Remarks to the Author:

I appreciate authors' effort in answering my earlier questions. Comparisons presented in the newly included Fig. S7 based on synthetic dataset considering system size enforces the self-consistency of the whole argument, and provides additional support of the photon bubble scenario. I recommend this work accepted for publication.

Answers to Reviewer 3

Comment 1: *I appreciate authors' effort in answering my earlier questions. Comparisons presented in the newly included Fig. S7 based on synthetic dataset considering system size enforces the self-consistency of the whole argument, and provides additional support of the photon bubble scenario. I recommend this work accepted for publication.*

REPLY: We thank the referee for his contribution: we are pleased with his recognition of our work.